# Public Baseline and shared response structures support the theory of antibody repertoire functional commonality

**Matthew I. J. Raybould**[1], **Claire Marks**[1], **Aleksandr Kovaltsuk**[1], **Alan P. Lewis**[2], **Jiye Shi**[3], **Charlotte M. Deane**[1] *

**1** Oxford Protein Informatics Group, Department of Statistics, University of Oxford, Oxford, United Kingdom, **2** Data and Computational Sciences, GlaxoSmithKline Research and Development, Stevenage, United Kingdom, **3** Chemistry Department, UCB Pharma, Slough, United Kingdom

* deane@stats.ox.ac.uk

**Data Availability Statement:** The four datasets released with this work are available from Zenodo at 10.5281/zenodo.4038176.

## Abstract

The naïve antibody/B-cell receptor (BCR) repertoires of different individuals ought to exhibit significant functional commonality, given that most pathogens trigger an effective antibody response to immunodominant epitopes. Sequence-based repertoire analysis has so far offered little evidence for this phenomenon. For example, a recent study estimated the number of shared ('public') antibody clonotypes in circulating baseline repertoires to be around 0.02% across ten unrelated individuals. However, to engage the same epitope, antibodies only require a similar binding site structure and the presence of key paratope interactions, which can occur even when their sequences are dissimilar. Here, we search for evidence of geometric similarity/convergence across human antibody repertoires. We first structurally profile naïve ('baseline') antibody diversity using snapshots from 41 unrelated individuals, predicting all modellable distinct structures within each repertoire. This analysis uncovers a high (much greater than random) degree of structural commonality. For instance, around 3% of distinct structures are common to the ten most diverse individual samples ('Public Baseline' structures). Our approach is the first computational method to find levels of BCR commonality commensurate with epitope immunodominance and could therefore be harnessed to find more genetically distant antibodies with same-epitope complementarity. We then apply the same structural profiling approach to repertoire snapshots from three individuals before and after flu vaccination, detecting a convergent structural drift indicative of recognising similar epitopes ('Public Response' structures). We show that Antibody Model Libraries derived from Public Baseline and Public Response structures represent a powerful geometric basis set of low-immunogenicity candidates exploitable for general or target-focused therapeutic antibody screening.

## Author summary

It is commonly thought that most people's adaptive immune systems can recognise the same endemic pathogens, many of which invade our bodies daily. However, existing

**Funding:** This work was supported by an Engineering and Physical Sciences Research Council (EPSRC) and Medical Research Council (MRC) grant [EP/L016044/1] awarded to MIJR, a Biotechnology and Biological Sciences Research Council (BBSRC) grant [BB/M011224/1] awarded to AK, and funding from GlaxoSmithKline plc, UCB Pharma Ltd., AstraZeneca plc, and F. Hoffmann-La Roche AG. EPSRC - https://epsrc.ukri.org/ MRC - https://mrc.ukri.org/ BBSRC - https://bbsrc.ukri.org/ GSK - https://www.gsk.com/en-gb/home/ UCB - https://www.ucb.com/ Roche - https://www.roche.com/ AstraZeneca - https://www.astrazeneca.co.uk/ GSK and UCB assisted in the analysis of the data presented in this manuscript.

**Competing interests:** I have read the journal's policy and the authors of this manuscript have the following competing interests. Alan P Lewis is employed by GSK, and Jiye Shi is employed by UCB. Both companies discover and sell antibody therapeutics. All other authors have declared that no competing interests exist.

methods of antibody repertoire comparison (which focus on genetic relatedness) only predict a tiny number of functionally equivalent antibodies in the resting state repertoires of different individuals. Here, we propose a novel approach that predicts the structural diversity of antibody binding sites within a repertoire sequence dataset. This orthogonal methodology can be applied to pool together antibodies from different genetic lineages with topological potential to bind to the same pathogen surface, and that may be functionally equivalent if they share a sufficiently similar surface interaction profile. Our methodology finds that a much greater than random set of binding site geometries exist across resting-state repertoires and can detect binding site geometric convergence in response to vaccination, both of which are consistent with underlying functional commonality between individuals. We further show that knowledge of these geometries could be useful in therapeutic antibody drug discovery, through rational screening library design. Different repertoire sequencing datasets could be interrogated to achieve a more general set of topologies compatible with many pathogens or a tailored set bespoke to a single pathogen.

## Introduction

A key component of the human immune system is the antibody/B-cell receptor (BCR) repertoire, a diverse array of immunoglobulins tasked with identifying pathogens and initiating the adaptive immune response. Broad pathogenic recognition is achieved through enormous variable domain sequence diversity, with an estimated $10^{10}$ unique heavy variable domains (VH) circulating at any one time from a theoretical set of $10^{12}$ (or $10^{16}$-$10^{18}$ full antibodies if light variable domain (VL) combinations are considered [1]).

On antigenic exposure, 'baseline' (resting-state) antibodies with sufficiently complementary binding sites to an antigen surface epitope are positively selected. The corresponding parent B cells subsequently migrate to the marginal zone of the lymph nodes, where intentional mutations are introduced to their sequence and only the highest-affinity binders survive in the competition for cognate T-helper cells [2].

Therefore, sequencing antibody repertoires before and during an immune response (e.g. vaccination) can reveal how different people respond to the same antigenic challenge, and can both improve our understanding of immunology and inform future vaccine or therapeutic design [3–5]. Similarly, comparing the repertoires of healthy individuals against immunosuppressed (*e.g.* HIV) patients may also make known the origins of increased disease susceptibility [6–8].

However, sequencing an entire antibody repertoire is challenging; they are so large that conventional sequencing techniques, such as Sanger sequencing, do not capture enough of the diversity to be informative. Instead, high-throughput immunoglobulin gene sequencing (Ig-seq) technologies (e.g. Illumina MiSeq) are used. These methods create snapshots that are typically on the order of $10^6$-$10^7$ VH and/or VL (unpaired) chains, up to a recent upper bound of around $10^9$ [1, 9, 10]. Single-cell sequencing methods, capable of preserving VH-VL chain pairings, are now emerging, however their current throughput yields datasets that are too small to study entire repertoire diversity [11–13].

Since most publicly-available Ig-seq data covers only the VH domain, the vast majority of whole-repertoire analysis has been performed over this region alone. The primary analytical method is currently 'clonotyping' [14–16]. Clonotyping is a computational technique used to sort sequencing datasets into sets of functionally similar chains based on sequence features, and can be performed in several ways. The most common implementation groups sequences

with the same predicted V and J gene transcript origins and above a certain percentage Complementarity-Determining Region H3 (CDRH3) sequence identity.

Such sequence-based approaches have contributed significantly to our knowledge of core immunology. For example, to estimate the true level of sequence similarity that exists across individuals, Briney et al. performed deep sequencing and clonotyping of the circulating baseline VH repertoires of ten volunteers [1]. They found that just 0.022% of observed clonotypes were 'public' (seen in everyone) and a similar study by Soto et al. found just ∼1% of clonotypes were public across three unrelated individuals. In a complementary approach, Greiff et al. trained a Support Vector Machine on public and private clonal sequences to identify their high-dimensional features, proving that they have distinct immunogenomic properties [17].

Clonotyping can also be used to detect antigen-specific immunoglobulins, through the identification of expanded clones after vaccination, or those present in unusually high proportions in individuals immune to certain diseases. Explorations of expanded lineages have yielded high-affinity antibodies and T cells against numerous pharmacologically interesting antigens, such as HIV proteins [6], cluster of differentiation proteins [18], botulinum neurotoxin serotype A [19], proteins implicated in type-1 diabetes [20], and many more.

However, clonotyping is only likely to identify a small subset of the true number of functionally equivalent antibodies. This is because it assumes that antibodies require a similar genetic background and high CDRH3 sequence identity to achieve complementarity to the same epitope. In reality, similar binding site structures and paratopes can be achieved from different genetic origins [21, 22] and with surprisingly low CDRH3 sequence identity [23] (conversely, false positives can arise where antibodies with high CDRH3 sequence identity and the same genetic origins adopt markedly different binding site topologies [23]). It is also the case that not every epitope is naturally suited to CDRH3-dominated binding, instead preferring broader engagement by multiple CDRs [24], putting less selection pressure on CDRH3 sequence identity.

It is difficult to reliably identify these hidden functionally equivalent antibodies within a clonotyping framework, as simply reducing the CDRH3 sequence identity threshold value lowers confidence in paratope residue similarity and increases the risk of grouping antibodies with fundamentally different binding site topologies. An alternative approach to relaxing the clustering criterion would be to initially ignore CDRH3 residue similarity, and instead to group antibodies with similar three-dimensional structures, as binders to a given epitope are likely to adopt a similar geometry. Geometrically-similar antibodies with sufficiently similar residue interaction profiles could then be capable of recapitulating key binding interactions at equivalent topological locations.

Experimental structure determination (e.g. by X-ray crystallography) remains too slow to solve representative portions of antibody repertoires [25]. However, structural annotation approaches are now fast enough to geometrically characterise the individual CDRs of millions of sequences a day with increasing accuracy [26, 27]. So far, these analyses have focused (consistency) solely on the VH chain, and none have considered the impact of VL on binding site configuration. This can most accurately be captured through variable domain (Fv) modelling, and recent developments have afforded homology approaches with sufficient throughput to analyse meaningful portions of the repertoire [28, 29]. For example, we developed a prototype structural profiling method that creates representative Fv model libraries from large repertoire snapshots, with applications in developability issue prediction [30].

In this paper, we further refine this repertoire structural profiler, and apply it to cluster antibody repertoires based on predicted binding site topology. We first analyse 41 naïve antibody repertoires from unrelated individuals, and find that the same representative ('distinct')

binding site structures are predicted to appear across many individuals ('Public Baseline' structures). We also show, through the construction of 'Random Repertoires', that this level of structural sharing is far greater than would be expected by chance. Our data therefore represents the first supporting computational evidence that considerably more functional commonality than suggested by clonotyping could exist in the baseline repertoires of different people. We then implement the same pipeline on pre- and post-vaccination datasets from three unrelated individuals, detecting a significant increase in structural commonality, and identifying all convergent response structures that may recognise similar epitopes ('Public Response' structures). We build Antibody Model Libraries (AMLs) by homology modelling a VH-VL sequence pairing predicted to adopt each Public Baseline or Public Response structure. *In silico* analysis of these AMLs suggests that they represent a powerful geometric basis set of low-immunogenicity candidates exploitable for general or target-focused therapeutic antibody screening.

## Results

This study comprises two main investigations. Firstly, we use data from an immunoglobulin gene sequencing (Ig-seq) study by Gidoni *et al.* [31] to investigate the degree of structural overlap in the circulating baseline repertoires of many unrelated individuals. We then use data from a longitudinal Ig-seq flu vaccination study by Gupta *et al.* [5] to measure three individuals' structural responses to exposure to a common antigen. Both translated Ig-seq datasets were downloaded from the Observed Antibody Space (OAS) database [9]. For the baseline repertoire study we retained only the 41 Gidoni volunteers with sufficient sequencing depth (see Methods).

We used an updated version of our Repertoire Structural Profiling pipeline [30] for improved accuracy in CDR structure and VH-VL interface orientation prediction (see Methods, S1 Text, S1–S4 Figs, and S1 Table). Briefly, Repertoire Structural Profiling takes as input an antibody/BCR repertoire snapshot containing heavy (VH) and light (VL) chain reads. It eliminates VH and VL chains for which not every CDR is modellable. All modellable VH and VL chains are then sequence clustered to reduce computational complexity. Surviving cluster centres are then paired together and the resulting Fvs that are likely to be successfully modelled are retained. Finally, predicted modellable Fvs with the same combinations of CDR lengths are structurally clustered based on the orientation and CDR loop templates forecast to be used during homology modelling. Antibody Model Libraries ('AMLs') can then be built from these representative Fv sequences.

### Structurally profiling the baseline immune repertoire

We first investigated the structural diversity present in the 41 selected Gidoni baseline repertoire datasets. Separately, each dataset was fed through our Repertoire Structural Profiling pipeline to identify the set of sequence diverse modellable VH and VL domains, then the number of predicted modellable Fvs, and finally the number of distinct structures in each dataset (Table 1, full table available as S2 Table).

The most structurally diverse dataset was 'S64' (209,394 distinct structures from $\sim$ 6.4M Fvs), and the least was 'S4' (78,588 distinct structures, from $\sim$ 750K Fvs). Datasets with a larger number of modellable sequence diverse VHs tended to result in a larger number of distinct structures. Datasets with a moderate/low number of modellable sequence diverse VHs but very large numbers of modellable sequence diverse VLs were amongst the least structurally diverse (*e.g.* 'S95'). This is consistent with our understanding of both length and structural variability in VH (particularly in CDRH3) relative to VL [32–34].

**Table 1. Structurally profiling the baseline repertoire snapshots [31].** A full table containing the values for all 41 baseline datasets is available in the Supporting Information (S2 Table). In order, the columns show: the dataset label, the number of VH and VL reads within each snapshot, the number of FREAD-modellable VH and VL reads (once clustered at 90% sequence identity), the number of predicted modellable Fvs resulting from these VH-VL pairings, and the number of distinct structures (cluster centres) identified in each dataset. Mod. = Modellable, SIC = Sequence Identity Clustered.

| Dataset | All VH | All VL | Mod. VH [90% SIC] | Mod. VL [90% SIC] | Predicted Mod. Fvs | Distinct Structures |
|---------|--------|--------|-------------------|-------------------|--------------------|--------------------|
| 1 (S64) | 177,603 | 123,934 | 10,087 | 6,779 | 6,420,211 | 209,394 |
| 2 (S57) | 169,805 | 118,020 | 9,860 | 7,922 | 7,225,630 | 201,039 |
| 3 (S5) | 159,544 | 139,845 | 8,999 | 8,526 | 6,827,419 | 200,708 |
| 4 (S56) | 162,446 | 136,874 | 9,309 | 7,168 | 6,628,683 | 195,061 |
| 5 (S83) | 152,299 | 112,733 | 9,048 | 8,076 | 6,170,373 | 193,384 |
| 6 (S67) | 173,722 | 120,237 | 9,349 | 6,424 | 5,544,952 | 193,061 |
| 7 (S84) | 164,017 | 138,874 | 8,702 | 8,232 | 5,634,598 | 191,617 |
| 8 (S76) | 148,180 | 126,713 | 8,778 | 7,047 | 5,856,150 | 191,162 |
| 9 (S54) | 121,993 | 133,921 | 7,581 | 9,066 | 5,074,822 | 181,290 |
| 10 (S89) | 152,710 | 144,340 | 8,923 | 9,293 | 5,414,820 | 177,829 |
| . . . | . . . | . . . | . . . | . . . | . . . | . . . |
| 39 (S95) | 118,576 | 162,377 | 5,412 | 11,748 | 5,901,443 | 91,855 |
| 40 (S17) | 102,405 | 111,669 | 5,310 | 7,945 | 2,690,081 | 91,229 |
| 41 (S4) | 100,689 | 128,986 | 4,688 | 1,761 | 745,977 | 78,588 |

## Expected numbers of distinct structures (*via.* 'Random Repertoires')

To contextualise the numbers of distinct structures observed for each baseline repertoire, we generated 'Random Repertoires' to obtain expected numbers of distinct structures assuming each genuine repertoire sampled randomly from modellable, accessible structure space. To achieve this, we derived:

(a) The *Modellable Repertoire Structures*: a sample of over 180 million structures built from a random combination of an orientation template, a CDR3 template, and a pair of CDR1/CDR2 templates from the same SAbDab entry (mimicking V gene-encoded predetermination). All CDR templates used had been previously assigned by FREAD to a human CDR. All Fv templates used had been previous assigned by interface residue comparison to a human VH-VL pairing.

(b) The *Length-Accessible Repertoire Structures* for each baseline snapshot: the subset of the Modellable Repertoire Structures with a CDR length combination observed in that individual.

(c) A *'Random Repertoire'* for each baseline snapshot: the appropriate Length-Accessible Repertoire Structures dataset was sampled the same number of times as that individual's number of predicted modellable Fvs. Clustering these 'Random Repertoires' then provided a reference number for the expected number of distinct structures per repertoire, given the depth of sampling in each dataset and assuming random sampling.

To derive a set of Modellable Repertoire Structures, we took the same number of samples as the number of Fvs derived from all baseline repertoire snapshots (183,544,740, S2 Table). Upon structural clustering, these samples yielded ∼24.4M distinct structures over ∼39.9K distinct combinations of CDR lengths, roughly 100x as many distinct structures as seen in any baseline repertoire sample. However, as each repertoire snapshot typically only contained between 2,000-3,500 different CDR length combinations, many of these 24.4M distinct structures could never be observed in the real data. Therefore, 41 'Length-Accessible Repertoire

**Table 2. Public structure analysis across the ten most structurally diverse baseline repertoire snapshots.** A table tracking the public structures across all datasets is available as S3 Table. A statistical estimate for the number of public structures was derived by randomly sub-sampling each Random Repertoire to the yield the same number of distinct structures (DSs) as its equivalent baseline repertoire snapshot. The 'Public Baseline' Antibody Model Library was derived from the 27,389 shared structures up to volunteer S89.

| # Repertoires Added | Fvs Added | Cumulative DSs | Public DSs (% Public) | Expected Public DSs (% Public) |
|---|---|---|---|---|
| 1 (S64) | 6,420,211 | 209,394 | 209,394 | 209,394 |
| 2 (+S57) | 7,225,630 | 340,915 | 100,824 (29.57%) | 12,307 (3.10%) |
| 3 (+S5) | 6,827,419 | 445,045 | 71,743 (16.12%) | 1,600 (0.28%) |
| 4 (+S56) | 6,628,683 | 527,668 | 58,043 (11.00%) | 322 (0.06%) |
| 5 (+S83) | 6,170,373 | 604,124 | 48,703 (8.06%) | 86 ($< 0.01\%$) |
| 6 (+S67) | 5,544,952 | 670,833 | 42,277 (6.30%) | 31 ($< 0.01\%$) |
| 7 (+S84) | 5,624,598 | 734,374 | 37,151 (5.06%) | 17 ($< 0.01\%$) |
| 8 (+S76) | 5,856,150 | 793,831 | 33,572 (4.23%) | 9 ($< 0.01\%$) |
| 9 (+S54) | 5,074,822 | 846,670 | 30,474 (3.60%) | 6 ($< 0.01\%$) |
| 10 (+S89) | 5,414,820 | 896,328 | 27,389 (3.06%) | 4 ($< 0.01\%$) |

Structures' datasets were created, limiting the Modellable Repertoire Structures to the CDR length combinations seen in each snapshot. For example, considering only the 3,468 CDR length combinations observed in our most structurally diverse individual ('S64') reduced the Modellable Repertoire Structures to a Length-Accessible Repertoire Structures dataset of $\sim$154.5M structures. This clustered into $\sim$18.0M distinct structures (a 26.2% reduction from the Modellable Repertoire Structures, while the number of CDR length combinations dropped $\sim$91.3%), implying we have good structural sampling over the CDR length combinations typically seen in humans. Every Length-Accessible Repertoire Structures dataset contained a number of randomly-selected structures roughly 20-30 times larger than the number of predicted modellable Fvs observed in the corresponding baseline repertoire.

Finally, 41 separate 'Random Repertoires' were created to determine the expected number of distinct structures assuming random structural sampling and given the observed structural sampling depth (see Methods). To do this, each individual's Length-Accessible Repertoire Structures were sampled randomly, without replacement, the same number of times as the number of predicted modellable Fvs (Table 2).

Again taking 'S64' as an example, the 6,420,211 samples comprising 'Random Repertoire S64' yielded 2,092,117 distinct structures, equating to an average of 3.07 Fvs per distinct structure, compared to 30.66 (9.99x more) Fvs per distinct structure in the genuine repertoire. This provides strong evidence that the modellable portions of antibody repertoires occupy a highly focused region of modellable structure space—roughly 10% of the expected number given the sample size (Fig 1), and 1% of a theoretical maximum estimate, across the same CDR length combinations.

## 'Public Baseline' structures in unrelated individuals

We next investigated whether structural commonality exists between baseline repertoire snapshots. This phenomenon would be statistically extremely unlikely by chance, given the focused structural sampling observed in each repertoire. To do this, we performed structural clustering on pairs of repertoire snapshots, looking for evidence of structural overlap (i.e. distinct structures assigned to a predicted modellable Fv seen in both datasets, see Methods and Fig 2).

Repertoire snapshots were ordered by their internal structural diversity ('S64' first, through to 'S4'). The 209,394 distinct structures of S64 act as a reference set of cluster centres. The 7,225,630 Fvs from snapshot S57 were then compared to these S64 cluster centres. Structures

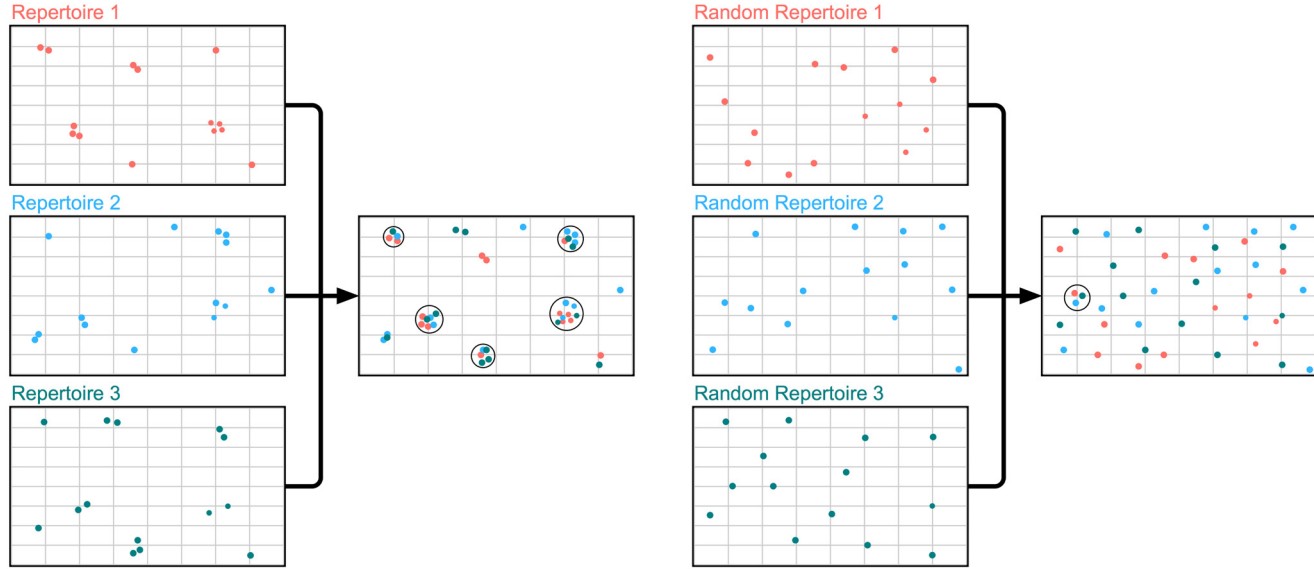

**Fig 1. Comparing genuine repertoire snapshots to synthetic 'Random Repertoires' (RRs).** Each dot represents a distinct structure mapped onto a two-dimensional representation of 'Length-Accessible Repertoire Structure' space. The genuine repertoire snapshots of all three individuals (red = repertoire 1, blue = repertoire 2, green = repertoire 3) exhibit focused structural sampling, covering ∼10% of the space as the corresponding RRs. Overlap analysis shows a high proportion of genuine repertoire distinct structures can characterise an Fv in all three individuals ('public structures', represented by black circles). When the same overlap analysis is performed on the equivalent 'Random Repertoires', far fewer public structures are observed.

present in both S57 and S64 were termed public across two individuals, while S64 and S57 distinct structures unique to their own dataset were termed private. Next, the 6,827,419 Fvs from S5 were compared to all public and private distinct structures observed in S64 and S57. We again evaluated the number of public structures, this time present in all three datasets. We

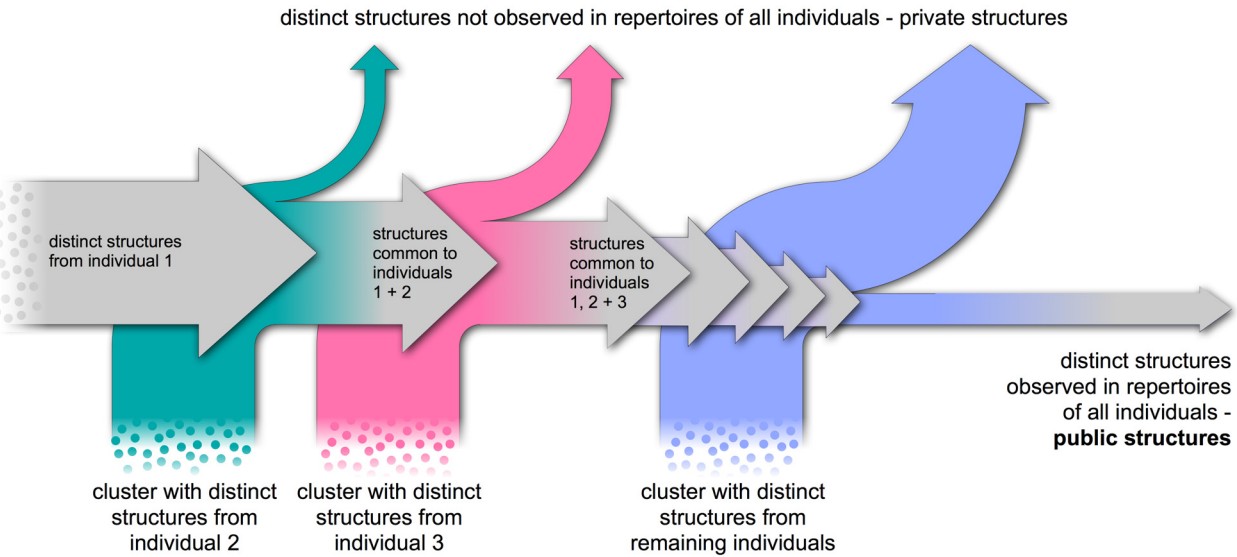

**Fig 2. Structural overlap analysis.** Datasets are arranged in order of their internal structural diversity (most diverse first). Distinct baseline structures from individual 1 are clustered sequentially with all other repertoire snapshots. Distinct structures present in every tested dataset are classed as 'public structures', whereas those that are absent in at least one individual are termed 'private structures'.

repeated this analysis for all remaining baseline repertoire snapshots (first ten results in Table 2, all 41 results in S3 Table).

To date, all *in silico* analysis of antibody repertoires has suggested that this number should drop rapidly towards 0. For example, a recent clonotype analysis of the baseline circulating repertoire estimated that only around 0.022% of clonotypes were public across ten unrelated individuals [31]. However, using our methodology, we found that the number of public distinct structures decreased at a far slower rate, still totalling 27,389 structures after ten unrelated individuals (Table 2). This represents 3.06% of all distinct structures observed up to that point, over 100 times the number of public clonotypes found by Briney *et al.* in their much deeper repertoire samples. Clonotyping our baseline snapshots, even at the lower 80% CDRH3 sequence identity threshold used by Soto et al. [35], revealed < 0.01% public clones after five individuals (S4 Table).

To provide a statistical estimate for how many distinct structures would be expected to be shared across these ten baseline repertoires, the Random Repertoire distinct structures were subsampled to match the corresponding number of baseline repertoire distinct structures (see Methods). In contrast to the genuine repertoires, the Random Repertoires overlapped sparsely, reaching ≤ 0.01% public structures by just the fifth volunteer (Table 2).

We also tracked the cumulative number of public and private structures over all 41 baseline repertoire snapshots (S5 Table). Even after the first few most diverse datasets, the deviation from an expected number of distinct structures (given the same ratio of distinct structures: modellable Fvs observed in S64) is quite substantial. This suggests that we might not expect much deviation from our observed fraction of public baseline distinct structures upon deeper repertoire sampling.

Finally, we tested whether the observed proportion of 'Public Baseline' structures would have been significantly different if the experiment had been run using an earlier FREAD database. We repeated Repertoire Structural Profiling for the two most structurally diverse datasets S64 and S57 removing any modellable Fv pairing whose best predicted template for any region was released by the PDB in 2018 or later. As expected, the number of predicted modellable Fv distinct structures in each sample fell from 209,394 and 201,039 to 186,677 and 179,763 respectively (a fall of around 10%). We then performed structural overlap analysis on these sets of distinct structures, finding a total of 305,948 distinct structures across both datasets, of which 87,920 were public to both S64 and S57. This degree of structural sharing (28.7%) is comparable to the degree observed with access to the entire FREAD database (29.6%).

The existence of so many 'Public Baseline' structures would be statistically extremely unlikely without the presence of underlying selection pressures promoting certain binding site topologies. Clonotyping, which conditions on sequence identity alone, has thus far been unable to detect significant similarities in the baseline repertoires of many individuals, even on much deeper sequencing samples. However, same-epitope complementarity ought to be governed by both structural and paratopic similarity, which may not correspond with conservation of gene transcript origin or high CDRH3 sequence identity. By relaxing the sequence identity criteria and instead focusing solely on geometric similarity, Repertoire Structural Profiling is the first computational method to provide supporting evidence for the levels of baseline antibody functional commonality implied by epitope immunodominance.

## Characterising the 'Public Baseline' structures

**CDR3 length usages.** We compared the North-defined [32] CDRH3, CDRL3 and CDRH3+CDRL3 distributions of the S64 Fv sequences assigned to a 'Public Baseline' structure against those assigned to a 'Private Baseline' structure (S5 Fig). The CDRL3 and

CDRH3+CDRL3 length usages demonstrate that 'Public Baseline' structures are not an artefact of using shorter CDR3 loops with more limited conformations. In fact, we find that modellability bias is likely to be overstating the proportion of 'Public Baseline' distinct structures with longer CDRH3 loop lengths. The structural space available to long CDRH3 (20+) loops is enormous, and we have relatively poor template structural coverage. As a result, if an Fv containing a long CDRH3 loop is considered modellable, it is more likely to be assigned to a structural template further away from its true structure, thus artificially inflating the numbers of long CDRH3s that look structurally similar. These longer CDR length 'Public Baseline' structures should therefore be treated with caution and, as more templates of longer CDRH3 loops emerge improving CDRH3 modellability, we would expect their numbers to decrease to the public:private ratios seen at more moderate CDRH3 lengths.

**Germline proximity and usages.**   We also investigated whether S64 Fv sequences assigned to 'Public Baseline' distinct structures were more proximal to germline than those assigned to 'Private Baseline' structures (S6 Fig, see Methods). The germline proximity of both 'Public' and 'Private' Fvs to their closest IGHV and IG[K/L]V genes is very similar, indicating that 'Public Baseline' structures are not solely an artefact of human V gene biases. Finally, we considered the constituent paired V genes across the 'Public Baseline' structures. As our pairing algorithm only predicts modellable Fv pairings based on PDB structures, we compared our IGHV/IG[K/L]V pairing frequencies with those observed in DeKosky *et al.*'s study of over 2000 natively-paired antibodies (S7 Fig) [11]. our 'Public Baseline' gene pairing frequencies were very similar to DeKosky *et al.*'s native sample, with the IGHV1/IGKV1-4, IGHV1/IGLV1-3, IGHV3/IGKV1, IGHV3/IGKV3, and IGHV3/IGLV1-4 pairings the most abundant.

**CDR template usages.**   We investigated the number of different structural templates that were assigned to each CDR in a 'Public Baseline' distinct structure (S6 Table). As expected, the lowest median number of different templates per distinct structure was recorded for the CDRH3 loop (2 templates/structure), consistent with the large structural variation within the region driving the definition of distinct binding site structures. Collectively, the light chain CDRs recorded considerably more FREAD templates per structure (median of 20 templates/structure) than the heavy chain CDRs (median of 9 templates/structure). We have supplied the sets of FREAD templates assigned to each CDR of each distinct structure to facilitate further structural characterisations of distinct structures of interest.

## Building and characterising a 'Public Baseline' antibody model library

We used ABodyBuilder [28] to construct an Antibody Model Library (AML) based on the 27,389 'S64' pairings predicted to adopt a 'Public Baseline' structure (as defined by the ten most structurally diverse repertoire snapshots). Some Fvs failed to be entirely homology modelled. For example, occasionally the CDRH3 template clashes irreparably with the CDRL3 template during construction of the full Fv model, necessitating *ab initio* treatment. Overall, 23,700 (86.53%) of 27,389 pairings were entirely homology modelled and comprise our 'Public Baseline AML'.

**Proximity to therapeutics.**   Predicted structures shared between many individuals might represent good starting points for therapeutic development. Their widespread nature could point to their binding versatility, and also to broad immune system tolerance across many individuals, lowering the risk of drug immunogenicity. To test whether our 'Public Baseline' AML contains antibody geometries proximal to known therapeutics, we mined Thera-SAbDab [36] for all 100% sequence identical structures of WHO-recognised therapeutics, selecting one per therapeutic (see Methods). Of the 66 therapeutics with known structures that had at least one antibody in our 'Public Baseline AML' with 6 identical CDR lengths, all had a structural

partner in the AML within a $C_\alpha$ Fv RMSD of 1.84Å, and 37 (56.1%) had a structural partner within 1.00Å Fv RMSD. Eleven therapeutic structures lay within 0.75Å Fv RMSD of a 'Public Baseline' AML structure (S7 Table); these therapeutics spanned a wide range of targets and were primarily successful or promising drugs (4 approved, 5 active in Phase III, 1 active in Phase II, and 2 discontinued).

This result demonstrates that the antibody models within our 'Public Baseline AML', without any explicit design, can display high levels of geometric similarity to known therapeutics. To show that similar binding site residue profiles can also be found by Repertoire Structural Profiling, we examined 'Public Baseline' distinct structure 'H14012+L14649' as a case study (Fig 3).

This structure lies within 0.64Å of the therapeutic Ustekinumab (S7 Table). Examining the backbone-aligned structures shows this difference lies in slightly different CDR loop structures assigned to the CDRH2, CDRH3, and CDRL3 loops (Fig 3A). We then examined all 4,911 Fv sequences assigned to this distinct structure across the ten individuals (S64 through S89), looking for the closest CDR sequence identity matches to Ustekinumab. The most similar of the 155 sequence-unique VH sequences assigned to this distinct structure is shown in Fig 3B. While both the Ustekinumab and 'Public Baseline' VH sequences most closely aligned to the same V and J genes (IGHV5-51/IGHJ4), the CDRH3 sequences are only 66% sequence identical, and so would not have been assigned to the same VH clonotype (the typical minimum threshold is 80% identity as used in Soto *et al.* [35]). This VH was observed coupled both with the VL sequence shown in Fig 3B and with the VL sequence shown in Fig 3C. The VL in Fig 3B is more identical across the three CDRs (22/26, 85%), while the one in Fig 3C is closer in CDRL3 identity but considerably less so in CDRL2 identity. Both these VLs derive from different IGKV germlines to the Ustekinumab VL (Ustekinumab: IGKV1D-16, Fig 3B VL: IGHV1-9, Fig 3C VL: IGKV3-15). Overall, the Fv described in Fig 3B is 75% sequence identical to Ustekinumab across all 6 CDRs.

This level of sequence and structural similarity between clinical-stage therapeutic antibodies and a representative of the 'Public Baseline' structural repertoire suggests that Repertoire Structural Profiling could prove an effective tool for designing general screening libraries containing promising drug leads.

**VH sequence profiling the 'H14012+L14649' distinct structure.** We performed clonotyping (80% sequence identity threshold [35]) on the 155 sequence non-redundant VH chains to determine the diversity of heavy chain clonotypes mapped to the 'H14012+L14649' Public Baseline structure. The VH sequences clustered into 141 distinct clonotypes, whose germline gene combinations as assigned by ANARCI [37] are shown in S8 Table. As clonotyping conditions on antibodies having the same V and J gene identities, it would never pool these VHs into a single category. Twelve of the 141 clonotypes have multiple occupancy (S9 Table). Three clonotypes were found across multiple individuals:

V5-51+ARPYGSGSYSDY+J4: seen in S64, S54, and S76

V5-51+ARQGYGDYVTDY+J4: seen in S67 and S76

V5-51+ARMGARPGYFDY+J4: seen in S89 and S76

This shows how Repertoire Structural Profiling could be used in conjunction with clonotyping to add geometric support to convergent clones being functionally equivalent. Recently published methods that can predict paratope similarity across all six CDRs [22, 38] may be able to find considerably more antibodies within each distinct structure cluster with similar enough interaction profiles to be functionally equivalent. To facilitate future investigations

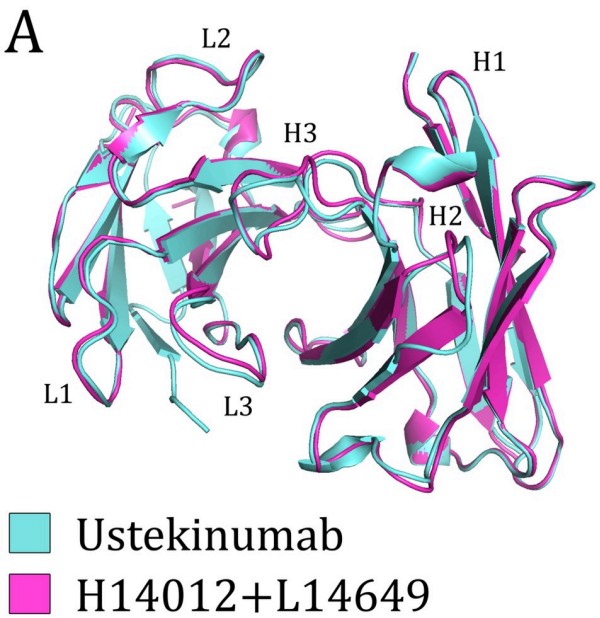

A

Ustekinumab

H14012+L14649

B

Ustekinumab, Fv assigned to H14012+L14649

```
VH: EVQLVQSGAEVKKPGESLKISCKGSGYSFTTYWLGWVRQMPGKGLDWIGIMSPVDSDIRYSPSFQGQV
VH: -VQLVQSGAEVKTRGESLKLSCKGSGYSFTSYWTGGVRQMPGKGLEWTGIIYPGDSDTSNSPSFQGQV
```

```
TMSVDKSITTAYLQWNSLKASDTAMYYCARRRPGQGYFDFWGQGTLVTVSS
TISADKSSSTGYLQWSSLKASDTAMYYCARREVGMGYFDYWGQGTLVTVSS
```

```
VL: DIQMTQSPSSLSASVGDRVTITCRASQGISSWLAWYQQKPEKAPKSLIYAASSLQSGVPSRFSGSGSG
VL: DIQLSQSPSFLSASVGDRVTITCRASQGISSSLAWYQQKQGKAPKVLIYAASTLQSGVPTRFRGSGSG
```

```
TDFTLTISSLQPEDFATYYCQQYNIYPYTFGQGTKLEIK
TEFTLTISSLQPEDFATYYCQQLNSCPPTFGQGTRLEIK
```

C

```
VL: DIQMTQSPSSLSASVGDRVTITCRASQGISSWLAWYQQKPEKAPKSLIYAASSLQSGVPSRFSGSGSG
VL: EIVMTQSPATLSVSPGARATLSCRASQSVNSNLAWYQQKPGQAPRLIIFGSSTRSTGIPVRFSGGGSG
```

```
TDFTLTISSLQPEDFATYYCQQYNIYPYTFGQGTKLEIK
TEFTLTISGLQSEDFEVYYCQQYNNWPYTFGQGTKLEIK
```

**Fig 3.** (A) Alignment of the solved Ustekinumab crystal structure (3hmw) and the closest Public Baseline AML structure (H14012 +L14649). (B) Comparison of the Ustekinumab Fv sequence and a Gidoni *et al.* naïve Fv sequence assigned by Repertoire Structural Profiling to the H14012+L14649 Public Baseline structure. The North-defined CDR regions of each chain are highlighted in bold. (C) An alternative VL sequence coupled to the same Gidoni VH sequence. This sequence has a more sequence similar CDRL3 but a less similar CDRL2.

into this area, we supply the Fv sequences across all ten individuals assigned to each 'Public Baseline' distinct structure.

## Structurally profiling a flu vaccine response

Clonotyping is commonly used in antibody drug discovery to identify 'expanded clones'—novel genetic lineages present after vaccination/infection but that were absent, or low concentration, beforehand [14]. Often these expanded lineages are seen across many different individuals after vaccination, implying particular pathogenic epitopes are 'immunodominant'—more susceptible to immune recognition [39–41]. Here, we applied Repertoire Structural Profiling to investigate whether we could identify an analogous public structural response to vaccination.

To this end, we used a longitudinal 2009 seasonal flu vaccination study by Gupta *et al.* [5], in which three unrelated individuals ('V1-3') were sequenced at many time-points before and after vaccination. Sequences were again downloaded from the OAS database, yielding 'Before Vaccination' and 'After Vaccination' datasets for each individual, according to the protocol described in the Methods. Using the same repertoire structural profiling protocol as above, we calculated the number of distinct structures observed in each individual before and after vaccination (S10 Table).

To obtain an estimate for the degree of structural commonality pre- and post-vaccination, we again used a greedy clustering approach to evaluate the structural overlap between the 'Before Vaccination' datasets, and between the 'After Vaccination' datasets, separately (Fig 4A and 4B). The first dataset in each overlap assessment was the most structurally diverse (*i.e.* the 'V3' individual before vaccination, and 'V1' after vaccination).

Again, a significant number of public distinct structures were observed in 'V1', 'V2', and 'V3' ('Public Before Vaccination' structures, 17.78% (236,792/1,444,597) of all 'Before Vaccination' distinct structures). This indicates that the identification of 'Public Baseline' structures in the previous section was unlikely due to serendipitous Ig-seq amplification bias. Interestingly, 17.78% is a similar percentage of sharing as that seen after three baseline snapshots (16.12%;

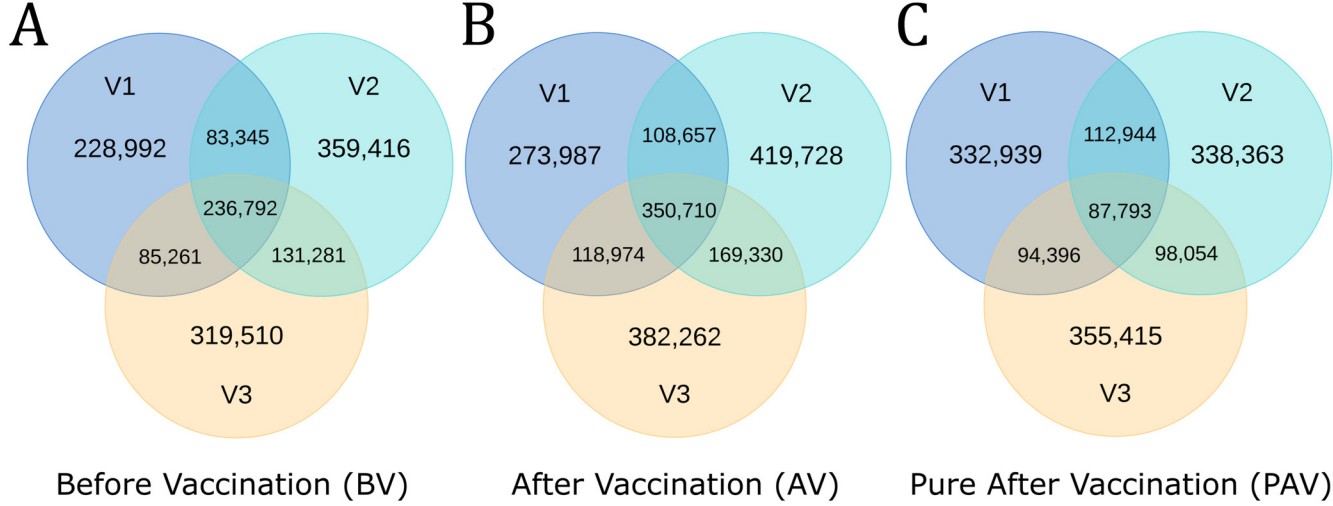

**Fig 4. Venn diagrams showing the structural overlap between each individual's A: 'Before Vaccination' dataset, B: 'After Vaccination' dataset, and C: 'Pure After Vaccination' dataset (distinct structures arising only after vaccination).** Total distinct structures: Before Vaccination—1,444,597; After Vaccination—1,823,628; Pure After Vaccination—1,419,904. V1-V3 = Volunteer 1-3.

71,743/445,045). For context, the proportion of all clonotypes that were public before vaccination was just 0.03% (Soto *et al*. definition [35], S11 Table).

The degree of structural sharing appears to increase after vaccination, with 19.23% (350,710/1,823,648) public structures across the three volunteers. This is consistent with a degree of repertoire structural convergence driven by exposure to the same pathogenic epitopes and with an increase in the proportion of public clonotypes after vaccination to 0.13% (S11 Table).

To derive these convergent structures, the structural overlap between each individual's 'Before Vaccination' and 'After Vaccination' datasets was measured, only retaining 'After Vaccination' pairings that could not be clustered into the same individual's 'Before Vaccination' distinct structures. 'V1' remained the most structurally diverse dataset, with 628,072 'Pure After Vaccination' distinct structures. The overlap between these 'Pure After Vaccination' pairings (Fig 4C) was then compared. This yielded a mixed picture of convergent and private vaccination response structures—27.7% (393,187/1,419,904) of distinct structures were shared with at least one other individual, and 6.18% (87,793/1,419,904) were shared across all three individuals—which we term 'Public Response' structures.

There are two potential causes of overlap in the 'Pure After' vaccination set. One is a genuine common structural response to vaccination, while the other is that the initial baseline repertoire was under-sampled—*i.e.* the overlap reflects residual shared baseline structures. As a second test for baseline deviation, beyond absence before vaccination, we compared how many of the 27,389 'Public Baseline' distinct structures were within 1Å of a 'Public Before Vaccination' binding site, versus the number within 1Å of a 'Public Response' Structure binding site. We observed that 80.0% (21,922/27,389) of 'Public Baseline' structures were within 1Å of a 'Public Before Vaccination' structure, compared to just 24.2% (6,621/27,389) proximal to a 'Public Response' structure. This provides further evidence that a proportion of these convergent 'Public Response' structures reside in a distinct region of structural space and could harbour epitope-specific binding geometries. We have built a 'Public Response AML' based on these 87,793 shared structures, with 74,181 Fvs (84.4%) entirely homology modelled.

## Discussion

In this work, we have structurally profiled antibody repertoires to capture new insights into the baseline and antigen-responding immune system, and to create novel libraries of public antibody structures that could be exploited for immunotherapeutic discovery.

All of the structural analysis in this paper is limited to the antibody chains that are currently predicted to be modellable, and so there remain regions of natural structural space uninvestigated and, once these become characterisable, the currently observed proportion of public structures may become diluted. Despite this, we show that antibody repertoires tend only to explore highly focused regions of currently-modellable structural space ($\sim$10% of the diversity expected if templates were explored randomly across the same combinations of CDR lengths). Coupled with our experiment blinding Repertoire Structural Profiling to the most recent year's templates, this suggests that a large portion of structural commonality will remain across the currently unmodellable regions of structural space (although we do expect the number of 'Public Baseline' structures with long CDRH3 loops to fall, as modellability may be increasing this figure).

The enormous sequence diversity exhibited across baseline antibody repertoires has long appeared to run contrary to the observation of baseline functional commonality—how are repertoires with such low clonal overlap able to respond in a timely manner to infection, usually to the same epitopes? Here we have shown that, at least from a structural perspective, there

is considerable opportunity for functional commonality across the circulating resting-state repertoires of unrelated individuals ($\sim$3% of observed distinct structures are public across 10 individuals). The theoretical chemical diversity that could be displayed on each of these scaffolds is large, so many of these grouped binding sites will not be complementary to the same antigen epitope. However, there is good reason to believe that a certain proportion are, as geometric similarity is a likely prerequisite of functional commonality, and our structural clustering approach offers a route to detecting and analysing these antibodies. We note that some edge cases remain in our analysis. It may be possible to identify structurally similar binding sites that use loops of different lengths through analysis of the resulting AMLs, but they are not readily detectable during this implementation of the clustering protocol. Antibodies that can use different CDRs to fit the same epitope *via* an alternative binding mode are also currently undetectable using our framework.

Once grouped into public structures, Fvs can then be probed using an array of methods designed to measure binding residue similarity to identify the subset likely to have common functionality. For example, finding convergent clonotypes within the public baseline structures may bolster confidence in their functionally convergent role. Alternatively, methods that do not condition on predicted antibody genetic origin, such as paratyping [22] or Ab-Ligity [38], could identify more genetically divergent antibodies capable of binding the same epitope. The public geometries themselves could also be harnessed in vaccinology, such as identifying an epitope targetable by a 'Public Baseline' structure which may lead to a more reliable and convergent response.

We hypothesise that human 'Public Baseline' structures are more likely to display low levels of human immunogenicity and be versatile binders. Building full three-dimensional variable domain models of these distinct structures (an Antibody Model Library) produced geometries that were very close to several approved and late-stage active therapeutic antibodies targeting diverse antigens. To chemically elaborate this 'Public Baseline' structural basis set, an *in silico* or phage display library on the order of $10^6$-$10^7$ sequence-unique human antibodies could be created from the many different Fv sequences predicted to adopt each public distinct structure. Mutations are likely required to optimise the affinity of a 'Public Baseline' antibody against a chosen epitope. If performed randomly, these mutations could negate the benefits of using natural antibody leads. However, tools such as Hu-mAb can distinguish human sequences from those of other organisms to extremely high accuracy [42]. Integrating these algorithms into affinity maturation pipelines to restrict mutations to those that do not decrease sequence humanness should help to preserve the low immunogenicity of 'Public Baseline' lead antibodies.

Target-focused screening libraries against immunodominant epitopes are commonly derived through sequence analysis of longitudinal Ig-seq studies that track the immune response of many individuals to the same antigen. We show that when our methodology is applied to a longitudinal flu vaccination case study, we detect a higher level of structural convergence, commensurate with response to similar epitopes on the same antigen. We can also derive a large number of 'Public Response' structures, with divergent structural characteristics from the baseline repertoire. These could contain useful binding site structures exploitable for antigen-specific library design, and the related antibodies may require less engineering than 'Public Baseline' candidates to achieve therapeutic levels of affinity.

Whilst ever we must artificially pair VH/VL sequencing datasets, we cannot conclusively prove that multiple individuals raised the same Fv binding site geometry in response to vaccination. This could soon be rectified with the advent of single-cell sequencing studies investigating vaccine response dynamics [43]. Repertoire Structural Profiling could readily be

applied to such data by skipping the combinatorial pairing step, which would be expected to improve both speed and accuracy.

There are also inevitable biases in structurally profiling human antibody repertoire data to suggest antibody leads for drug discovery. One such biased property is CDRH3 length: very short CDRH3 lengths will be under-sampled through their sparsity in natural human sequences [30], while very long CDRH3 lengths will be under-sampled because they are more difficult to homology model accurately. While inherent immunogenicity should be diminished by virtue of using naturally-expressed sequences, other developability issues are still possible, as not every human antibody has the biophysical properties ideal for large-scale manufacture and long-term storage [30].

Nevertheless, we believe that our approach should be applicable both for designing *in silico/ in vitro* screening libraries and in assisting antibody functional annotation. We have made available the 'Public Baseline' and 'Public Response' Antibody Model Libraries for further investigation, and will continue to build and share the Antibody Model Libraries derived from other unpaired and paired VH+VL datasets in the Observed Antibody Space database [9].

## Methods

### Immunoglobulin gene sequencing datasets

The cleaned and translated antibody repertoire datasets [5, 31] were downloaded directly from the Observed Antibody Space (OAS) database [9]. For the Gidoni data [31], only individuals for whom > 100,000 IgM VH and >100,000 VL sequences were recorded were analysed. In our analysis of Gupta et al. [5], we used all three individuals ('V1' = 'FV', 'V2' = 'GMC', and 'V3' = 'IB'). The 'Before Vaccination' data was defined as all VH and VL sequences recorded at 8 days, 2 days and 1 hour before vaccination. The 'After Vaccination' data was defined as all VH and VL sequences recorded at 1 week, 2 weeks, 3 weeks, and 4 weeks after vaccination. Sequences recorded 1 hour and 1 day after vaccination were discarded to avoid ambiguity. The 'Pure After Vaccination' data contained 'After Vaccination' sequences that did not fall into the structural clusters defined by each individual's 'Before Vaccination' repertoires. The seminal work in which 'FV', 'GMC', and 'IB' were vaccinated is detailed in Laserson *et al.* [4], however the data we use derives from Gupta *et al.* [5], who re-analysed each antibody repertoire snapshot with Illumina sequencing.

### Repertoire structural profiling pipeline

The described structural profiling pipeline was optimised from the protocol reported in the Supporting Information of Proc. Natl. Acad. Sci. (2019) 110(6):4025-4030 [30].

**CDR modellability analysis.** Each sequence was first numbered using ANARCI [37] according to the IMGT numbering scheme [44], and the closest framework region (variable domain with North-defined CDRs [32] excised) in the SAbDab [24] database (12$^{th}$ February 2019) was identified. In the IMGT numbering scheme, the North CDRs lie between the following residue numbers—CDRH1: 24-40; CDRH2: 55-66; CDRH3: 105-117; CDRL1: 24-40; CDRL2: 55-69; CDRL3: 105-117.

FREAD [45, 46] was then used to attempt to map each Ig-seq sequence to a length-matched North CDR template. The FREAD CDR databases were timestamped to 12$^{th}$ February 2019, and contained the following numbers of templates—CDRH1: 2,526; CDRH2: 2,575; CDRH3: 2,502; CDRL1: 2,355; CDRL2: 2,373; CDRL3: 2,376. Templates were not restricted only to those with "human" PDB organism assignments for multiple reasons. Antibodies in the PDB are highly engineered, both through point residue mutations and entire loop transplantation, meaning single organism origin labels are only accurate for a small number of entries. In addition,

internal benchmarking of FREAD [45, 46] and ABodyBuilder [28] showed that including "non-human" templates in our FREAD loop databases (particularly the CDRH3 database) leads to greater structural coverage and a net improvement in CDR structure prediction accuracy. All loop templates contained the North-defined CDR loop and 5 'anchor residues' before and after the loop. Selection of CDRH3 templates was performed according to a bespoke set of Environment-Specific Substitution (ESS) score thresholds established using Ig-seq data: Lengths 5-8, ESS $\geq$ 25; Lengths 9-10, ESS $\geq$ 35; Lengths 11+, ESS $\geq$ 40 (see S1 Text). Each template surpassing the threshold was subsequently grafted onto the corresponding framework anchor residues. The loop template with the lowest calculated $C_\alpha$ anchor RMSD was selected. Any sequences for which at least one loop could not be modelled were removed from the dataset.

**Sequence clustering.** The modellable chains were then sequence clustered using CD-HIT [47] at a 90% sequence identity threshold, to reduce the number of VH-VL pairing comparisons to a computationally-tractable number.

**Predicting modellable VH-VL orientations.** The 20 most important VH-VL interface residues for orientation prediction were derived; a sequence identity of 85% over these 20 residues resulted in an orientation RMSD of $\leq 1.5$Å $\sim 80\%$ of the time (see S1 Text).

All remaining VH and VL domains after sequence clustering were paired together, and their 20 key interface residues were recorded. If the sequence identity over these residues was $\geq 85\%$ to at least one of 1,129 reference Fvs, the interface was classed as modellable—otherwise the VH-VL pairing was discarded. If multiple reference Fvs shared $\geq 85\%$ identity, the predicted modellable VH-VL pairing inherited the orientation parameters of the Fv reference with highest sequence identity.

**Identifying distinct structures.** At this stage, each predicted modellable VH-VL pairing (Fv) has eight associated parameters: its orientation template, its six CDR templates, and a length vector recording the combination of North CDR lengths [32] present in its binding site. Fvs were then structurally clustered to identify 'distinct structures' according to the following process. First, identically-predicted binding sites (for which the eight predicted parameters were the same) were identified. The retained pairing was randomly chosen, except in the overlap studies—if one of the pairings was present as a distinct structure of the first dataset, this pairing was selected and recorded as a shared structure across both repertoires.

Next, singleton length clusters were identified and assigned as separate distinct structures, avoiding inaccurate RMSD comparisons between loops of differing length. The remaining interfaces were split by their CDR length combinations, and were greedily clustered with all other pairings sharing the same length vector as follows:

1. Select the first pairing as a distinct structure (cluster centre).

2. Select the next pairing. If the orientation RMSD to all existing cluster centre orientation templates exceeds 1.5 Å, classify the new pairing as a distinct structure. Otherwise:

3. Calculate the RMSD between the CDR templates of the new pairing with those of all existing cluster centres using the formula:

$$\sqrt{\frac{\sum_X^{(H1-H3,L1-L3)} D_{X_{12}}^2 LX}{\sum_X^{(H1-H3,L1-L3)} LX}}$$

where the sum over X refers to each of the six CDRs, $L_X$ is the length of North CDRX, and $D_{X_{12}}$ is the $C_\alpha$ RMSD between the CDRX in Fv 1 and Fv 2. If this value exceeds 1 Å to all existing structural cluster centres, the pairing is assigned as a distinct structure. Otherwise the pairing is stripped from the dataset.

4.  Return to step 2 until all pairings have been analysed.

## Overlap comparison

To identify shared structures between two Ig-seq repertoire snapshots, the distinct structures from the first snapshot were listed followed by all predicted modellable Fvs of the second repertoire snapshot, as an input file to the clustering algorithm. The greedy clustering ensured that all distinct structures from the first dataset remained as cluster centres, and allowed for the identification of pairings in the second dataset that were also predicted to occupy the same structural neighbourhood.

## 'Random Repertoires'

To contextualise the structural diversity displayed in human antibody repertoires, we derived 'Random Repertoires' (RRs) according to the following method. First, a set of Modellable Repertoire Structures (MRS) was generated. When generating a structure, one of any of 663 orientation templates, 2,051 CDRH3 templates, and 2,125 CDRL3 templates previously assigned by FREAD to a human Fv/CDR sequence were available for selection. To mirror the genetics of the immune system (as they would be encoded on the same V gene transcript), CDR1 and CDR2 templates were restricted to being selected from the same SAbDab structure, limiting our choice to one of 789 CDRH1/2 templates and 912 CDRL1/2 templates, again all of which FREAD had previously assigned to human sequences. Each of these five sets was randomly sampled over 180 million times to create the MRS dataset. This was then filtered to create 41 Length-Accessible Repertoire Structure (LARS) datasets, containing only the combinations of CDR lengths observed in each baseline repertoire snapshot. Finally, RRs were created by sampling each LARS set the same number of times as the number of predicted modellable Fvs in the corresponding baseline repertoire snapshot.

To obtain statistically expected values for structural overlap across individuals, the distinct structures from 'RR_S64' were randomly subsampled the same number of times as the number of distinct structures seen in 'S64' itself, yielding random distinct structure samples occupying the same proportion of LARS-space. The process was repeated for each RR dataset, normalising to each respective baseline repertoire snapshot. Overlap comparison was then performed as described above, starting from the 'RR_S64' distinct structures, followed by all the pairings that fell into the selected 'RR_S57' distinct structures, *etc*.

## Clonotyping

Clonotyping was performed to group antibodies with the same closest V and J gene, and either identical CDRH3 sequences, as in Briney *et al.* [1], or with CDRH3 sequences within 80% sequence identity, as in Soto *et al.* [35].

## Antibody model library construction

Antibody model libraries (AMLs) were constructed with a parallel implementation of ABody-Builder [28], using the FREAD [45, 46] Environment Specific Substitution Scores derived from Ig-seq benchmarking (see CDR Modellability Analysis). Some predicted modellable Fvs are not entirely homology modellable, as loop modellability is considered on a per-chain basis and does not take into account inter-chain loop clashes that become evident only upon full Fv homology modeling. Fvs that required any degree of *ab initio* modelling to fix such issues were trimmed out of the dataset.

### Structural comparison to antibody therapeutics

The set of 89 therapeutics with 100% sequence identical structures (as of November 2019) were retrieved from Thera-SAbDab [36]. A single structure was chosen for each therapeutic for the RMSD analysis; if multiple structures were available, we selected unbound structures with the best resolution. RMSD comparisons were only made between therapeutics and AML structures with matching combinations of CDR lengths. Fv RMSD was calculated over all $C_\alpha$ atoms after alignment of backbone atoms, using an in-house script.

## Supporting information

**S1 Text. Supporting information methods.** A description of the methodology used to benchmark new ESS thresholds for use on repertoire data, and for evaluating a set of 20 important interface residues for orientation template assignment.
(PDF)

**S1 Fig. ESS benchmarking.** The percentage of each FREAD top-ranked CDRH3 templates with an Environment Specific Substitution Score (ESS) within the labelled bin for (a) a typical Ig-seq dataset, and (b) the Protein Data Bank (blinded to self). The two sets have very different distributions; notably Ig-seq datasets rarely contain CDRH3 loops with extremely high ESS scores to dataset templates.
(PNG)

**S2 Fig. Orientation variation for identical Fvs.** The distribution of orientation RMSDs observed between Fvs of identical heavy and light chain sequence. The vast majority (92%) have orientation RMSDs below 1.5Å
(PNG)

**S3 Fig. Orientation RMSD by VH-VL interface identity.** Graphs showing the orientation RMSD observed at each interface sequence identity value for (A) all 52 interface residues and (b) the 20 most important interface residues. The thresholds for (A) are set at 1.5Å and 82% sequence identity, while for (B) are set at 1.5Å and 85% sequence identity. The proportions above the sequence identity threshold and within 1.5Å orientation RMSD are 80.2% (982/1224) and 77.8% (954/1227) respectively.
(PNG)

**S4 Fig. The Repertoire Structural Profiling algorithm.** Heavy (VH) and light (VL) chain sequences from a repertoire snapshot are first analysed separately for their FREAD modellability (unmodellable chains are crossed out). They are then clustered by sequence identity using CD-HIT (90% threshold) for computational tractability. All VH and VL cluster centre chains are subsequently paired, and VH-VL orientations that cannot reliably modelled are removed (again shown by crosses). Finally, predicted modellable Fvs with identical combinations of CDR lengths are structurally clustered to identify 'distinct structures'.
(PNG)

**S5 Fig. CDR length distributions for S64 antibodies assigned to 'Public' vs. 'Private' structures.** Bar charts comparing the (A) CDRH3 lengths, (B) CDRL3 lengths, (C) Combined CDRH3+CDRL3 lengths of S64 sequences assigned to 'Public Baseline' structures (blue) against those assigned to 'Private Baseline' structures (orange).
(PNG)

**S6 Fig. Germline distributions for S64 antibodies assigned to 'Public' vs. 'Private' structures.** Histograms comparing the (A) closest IGHV germline sequence identity, and (B) closest

IGKV/IGLV germline sequence identity of S64 sequences assigned to 'Public Baseline' structures (blue) against those assigned to 'Private Baseline' structures (orange).
(PNG)

**S7 Fig. Germline family pairings in the 'Public Baseline' AML.** A heatmap showing IGHV: IGKV/IGLV gene family pairings across the 'Public Baseline' structures. The usage trends are consistent with the natural pairings observed in DeKosky *et al.* [11].
(PNG)

**S1 Table. VH-VL interface residues.** The 52 heavy and light chain residues tending to lie in the heavy-light chain interface. Residue numbers in bold were determined to be amongst the five most important in the Random Forest regression model when predicting the six different ABangle parameters.
(PNG)

**S2 Table. Applying Repertoire Structural Profiling to baseline repertoire samples.** Structurally profiling the baseline repertoire snapshots of 41 unrelated individuals. In order, the columns show: the dataset label, the number of VH and VL reads within each snapshot, the number of FREAD-modellable VH and VL reads (once clustered at 90% sequence identity), the number of predicted modellable Fvs resulting from these VH-VL pairings, and the number of distinct structures (cluster centres) identified in each dataset. SIC = Sequence Identity Clustered.
(PNG)

**S3 Table. Evaluating the number of 'Public Baseline' distinct structures.** Evaluating the number of public distinct structures seen across multiple baseline repertoire snapshots. In order, the columns show: the number of repertoires compared (in brackets the identifier of the last dataset added), the number of predicted modellable Fvs added by the last dataset, the number of distinct structures added by the last dataset, the (cumulative) number of public and private distinct structures across all compared repertoires, and the number of proportion of these structures that are public. The sharp drop-off in the proportion of public structures in the final four repertoire snapshots can be rationalised by their substantially lower internal structural diversity (see Table 2).
(PNG)

**S4 Table. Baseline repertoire shared clonotypes.** Tracking the number of public clonotypes shared across all naïve baseline datasets analysed up to that point (e.g. 358 clonotypes are present in S64, S57, and S5 according to the Soto V3J definition).
(PNG)

**S5 Table. Cumulative baseline repertoire structures identified.** Tracking the total number of public and private distinct structures seen across multiple baseline repertoire snapshots. In order, the columns show: the number of repertoires compared (in brackets the identifier of the last dataset added), the cumulative number of predicted modellable Fvs, the number of public and private distinct structures seen across all compared repertoires, and the expected number of cumulative public and private distinct structures if new distinct structures were observed at the same rate per modellable Fv as seen in S64.
(PNG)

**S6 Table. FREAD templates per 'Public Baseline' distinct structure.** The median numbers of unique FREAD templates assigned to each CDR within a 'Public Baseline' distinct structure.
(PNG)

**S7 Table. Structural comparison of 'Public Baseline' AML to clinical-stage therapeutics.**
The eleven clinical-stage therapeutic antibodies with a solved crystal structure within 0.75Å
variable domain (Fv) root-mean-squared deviation (RMSD) of an antibody model structure
from the Public Baseline Antibody Model Library (PB AML). The first column records the Fv
identifier for the geometrically closest AML model to each of the eleven therapeutics listed in
column 2. Column 3 provides the Protein Data Bank (PDB) identifier for each chosen thera-
peutic structure (chain identifiers in brackets). The corresponding RMSD is provided in col-
umn 4; all RMSD comparisons were made between AML structures and therapeutics with an
identical combination of CDR lengths. This combination of North-defined CDR lengths is
then listed in the order H1-H2-H3-L1-L2-L3. Finally, the target for each therapeutic
antibody is recorded. PDB = Protein Data Bank; VH = variable heavy chain; VL = variable
light chain; Fv = Fragment variable region; RMSD = root-mean-squared deviation;
CDR = Complementarity-Determining Region. Antigens: CD—Cluster of Differentiation pro-
tein, NGFB—Nerve Growth Factor B, IL—interleukin, TSLP—Thymic Stromal Lymphopoie-
tin, APP—Amyloid Precursor Protein, MIF—Macrophage Migration Inhibitory Factor, IGHE
—Immunoglobulin Heavy Constant Epsilon.
(PNG)

**S8 Table. Different clonotypes are mapped to the same distinct structure.** The diversity of
IGHV/IGHJ gene combinations represented across the 141 VH clonotypes assigned by Reper-
toire Structural Profiling to the 'H14012+L14649' 'Public Baseline' distinct structure.
(PNG)

**S9 Table. Multiple occupancy clonotypes assigned to the same distinct structure.** The 12
multiple-occupancy VH clonotypes assigned by Repertoire Structural Profiling to the 'H14012
+L14649' 'Public Baseline' distinct structure.
(PNG)

**S10 Table. Applying Repertoire Structural Profiling to baseline repertoire samples.** Struc-
turally profiling the 'Before Vaccination' (Before) and 'After Vaccination' (After) repertoire
snapshots of three unrelated individuals (V1, V2, and V3). In order, the columns show: the
dataset label, the number of VH and VL reads within each snapshot, the number of FREAD-
modellable VH and VL reads (once clustered at 90% sequence identity), the number of pre-
dicted-modellable Fvs resulting from these VH-VL pairings, and the number of distinct struc-
tures (cluster centres) identified through greedy structural clustering. SIC = Sequence Identity
Clustered.
(PNG)

**S11 Table. Flu vaccination repertoire shared clonotypes.** Tracking the number of public clo-
notypes shared across all "Before Vaccination" (Before) datasets and all "After Vaccination"
(After) analysed up to that point (e.g. 272 clonotypes are public across V1, V2, and V3 accord-
ing to the Soto V3J definition). The Briney definition clusters CDRH3s at 100% sequence iden-
tity and same V/J genes, while the Soto Definition clusters CDRH3s at 80% sequence identity
and same V/J genes.
(PNG)

## Author Contributions

**Conceptualization:** Matthew I. J. Raybould, Charlotte M. Deane.

**Data curation:** Matthew I. J. Raybould.

**Formal analysis:** Matthew I. J. Raybould.

**Funding acquisition:** Charlotte M. Deane.

**Investigation:** Matthew I. J. Raybould.

**Methodology:** Matthew I. J. Raybould, Claire Marks, Charlotte M. Deane.

**Project administration:** Charlotte M. Deane.

**Resources:** Charlotte M. Deane.

**Software:** Matthew I. J. Raybould.

**Supervision:** Claire Marks, Alan P. Lewis, Jiye Shi, Charlotte M. Deane.

**Visualization:** Matthew I. J. Raybould, Claire Marks.

**Writing – original draft:** Matthew I. J. Raybould.

**Writing – review & editing:** Claire Marks, Aleksandr Kovaltsuk, Alan P. Lewis, Jiye Shi, Charlotte M. Deane.

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
