## [Decision Letter · Decision Letter 0]

17 Jul 2020

Dear Prof Deane,

Thank you very much for submitting your manuscript "Evidence of Antibody Repertoire Functional Convergence through Public Baseline and Shared Response Structures" for consideration at PLOS Computational Biology.

As with all papers reviewed by the journal, your manuscript was reviewed by members of the editorial board and by several independent reviewers. In light of the reviews (below this email), we would like to invite the resubmission of a significantly-revised version that takes into account the reviewers' comments.

I apologize for the delay. I was waiting for an additional review since the reviewers had somewhat varied opinions on the manuscript. While the reviewers all appreciated the goals of the study and the significant efforts employed in the analysis presented in the paper, they share some misgivings on the interpretation of the results and whether the results could be an artifact of the necessary filters and cutoffs employed in defining "modellable" structures and the method for determining a random repertoire.

I have similar concerns. In particular, I wonder if the random repertoire is affected by not filtering the templates for human structures of CDRH1/2, CDRL1/2, CDRH3, and CDRL3. I don't understand the utility of including non-human structures in generating the random repertoire. I also wonder if the 3% Public Antibodies are closer to germlines than the private sequences, and may be overrepresented in the PDB. Some characterization of the properties of the 3% Public Antibodies may help to clarify this issue. Finally, because the study is performed on non-paired VH and VL sequences, it is difficult to see how it is possible to demonstrate that vaccination results in antibodies that bind to the same epitopes in the vaccine in the same way (i.e, roughly they would have very similar antigen/antibody co-complex structures such that the paratope/epitope interface and orientation are similar). Are there any paired VH/VL antibody sequence sets that could be added to the analysis?

We cannot make any decision about publication until we have seen the revised manuscript and your response to the reviewers' comments. Your revised manuscript is also likely to be sent to reviewers for further evaluation.

Sincerely,

Roland L. Dunbrack Jr., Ph.D.

Associate Editor

PLOS Computational Biology

Feilim Mac Gabhann

Editor-in-Chief

PLOS Computational Biology

Reviewer's Responses to Questions

**Comments to the Authors:**

Reviewer #1: Raybould et al. present a new approach to assess the degree to which antibodies with common modes of function exist across individuals in a population. The authors note that, to date, sequence-based analyses have not directly explained the broad, consistent immunogenic response to pathogens in a population, and suggest using structural comparisons to estimate functional commonalities that escape analyses based on sequence alone. Their approach first reduces the number of VH and VL sequences in an individual’s sequenced repertoire by clustering sequences grouped by shared framework regions and with the same CDR lengths. Templates for each CDR are grafted onto the framework, and the VH–VL orientation was modelled by examining 20 predictive residues. Any sequences that could not be modelled by this process are not considered. After models are constructed, they are clustered to identify groups of similar structures. The sets of models from individuals’ repertoires are then compared to identify structurally similar antibodies shared among individuals, termed public distinct structures. The authors find that approximately 3% of the structures are public, compared to 0.022% of sequences that can be classified as public through clonotype analysis. The authors then apply this analysis to a set of pre- and post-immunization repertoires to identify structurally similar antibody responses across three individuals.

This an interesting approach for exploring the frequency of groups of canonical CDRs and frameworks appearing together within and among individuals, but I do not believe the evidence, as reported, supports the authors’ objective or specific claims. In consideration of this along with the contextualization of the method with respect to the other work in the field, I do not feel this work meets the high standards for publication in PLOS Computational Biology.

In the abstract the authors state, “The antibody repertoires of different individuals ought to exhibit significant functional commonality, given that most pathogens trigger a successful immune response in most people. Sequence-based approaches have so far offered little evidence for this phenomenon. For example, a recent study estimated the number of shared (‘public’) antibody clonotypes in circulating baseline repertoires to be around 0.02% across ten unrelated individuals. However, to engage the same epitope, antibodies only require a similar binding site structure and the presence of key paratope interactions, which can occur even when their sequences are dissimilar.”

This raises a few questions that must be addressed: (1) Why would one expect that individual immune responses would all target the same epitope? Indeed, polyclonal responses demonstrate that many epitopes can be the target of an immune response; (2) Clonotyping has a much higher threshold for labeling a sequence “public” because it is attempting to capture sequence convergence as opposed to epitope-binding or paratope convergence. While two distinct sequences can bind the same epitope while not being classified as public via clonotyping, two antibodies having similar backbone conformations (and thus would be classified as sharing a public structure) does not imply that they bind the same epitope or even the same antigen. How can comparisons of these concepts be made such that they reveal an immunological truth?; (3) Why does having similar backbone conformations suggest that key paratope interactions are maintained? One can imagine two distinct antibodies (by essentially any definition) that could have nearly identical paratope–epitope interactions because those depend on the constellation of functional groups and hydrophobic surfaces arranged in space (i.e. it is possible to change the CDR loop lengths and conformations while maintaining the key interactions that give rise to an antibody’s affinity and specificity).

The authors argue that clonotyping is poorly suited to the task of identifying functionally similar antibodies “because it assumes that antibodies require a similar genetic background and high CDRH3 sequence identity to achieve complementarity to the same epitope. In reality, similar binding site structures and paratopes can be achieved from different genetic origins (21) and with surprisingly low CDRH3 sequence identity (22).” However, they do not provide evidence that the structure-based approach employed in this study would identify genetically diverse antibodies that are functionally similar across individuals. Unfortunately, I do not believe this structure-based approach can capture the kind of functional convergence required to shed light on the phenomenon under study.

Reviewer #2: Attachment

Reviewer #3: The work by Raubould et al is very interesting and important in highlighting the importance of structural knowledge of the antibody repertoire functional commonality.

The work is very well performed and interesting to read.

I have one major query that I think can help in validating the message further and better understand the role of the modelling in the final outcome.

Could this extent of 'public' (shared) antibodies be the artefact of the modelling - i.e. only could consider the modellable part of the repertoire (with templates etc.) which impose some restrictions on the CDRs and hence you would naturally find some recurring patterns of the CDRs shared across different individuals?

In other words how can one be sure that the modelling methods do not introduce bias that will increase the "public" sequences?

Also it is interesting to know which specific features of the antibody distinct structure elements contribute most to the public response structures and specifically are contributing to the observed 3% distinct structures in common between the studied individuals' repertoire.

Reviewer #4: The paper "Evidence of Antibody Repertoire Functional Convergence through Public Baseline and Shared Response Structures" studies the structural similarity of antibody from repertoires of different individuals, with the aim of identifying public structures, and to study the progression of immune responses.

The aim is important, but I have some concerns on some of the methods. In general, I believe that the paper should tone down its results and discuss the bottlenecks and critical points of the analysis

Major

- All the analyses are performed on bulk sequencing data, where VH and VL sequences are not paired, by pairing all VHs to all VLs. This means that most of the antibody studied are, in the best case, potentially observable but not present in the repertoire, and in the worst case, impossible to be produced. As an example following the paper: long H3 and L3 are rarely seen together. This means, for example, that pairing short H3 with short L3 antibodies would generate a subset of structures with limited diversity, but that are not necessarily real. This point has to be discussed and addressed. For example: is there a bias in the CDR length in public structures with respect of non public ones?

- The paper claims that the antibody structural space is far from a random selection, and that this hints that structural similarity might be fundamental to identify antigenic pressure in a better way than one can obtain by performing sequence-based analyses. While this makes sense, I believe that the randomisation tests performed fail to prove so. All the randomisation tests are performed by uniform sampling from a large pool of potential models with reasonable CDR3 lengths, coming from all possible germlines, both human and murine. So, the tests performed say that the structures in the repertoire are not generated by randomly sampling such a diverse dataset, and this is correct, bot not overly interesting. The repertoire of each individual is only generated by a non uniform selection from a small pool of genes (and only human of course). Moreover, many structures in the template database might be heavily engineered or contain a large amount of somatic mutations, so that they will be quite diverse from the probably nearly naive repertoires.

The authors should perform additional randomisation tests to show that there is a structural selection and that this might be due to antigenic pressure. as a first step, the randomised dataset could be generated by having the same distribution of CDR lengths and the same structural templates used in the actual repertoires. They could also generate artificial sequences (e.g. with IgOR, see "High-throughput immune repertoire analysis with IGoR" ). If shared structures are observed in such artificial repertoires also, this would mean that these are due to the genetic set-up of Igs, and not to antigenic pressure

- The amount of "modellable" Igs is just a small fraction of all the sequences, specifically those for which proper templates are available. This means that the high level of similarity might be caused by a selection bias, in conjunction with similarity by proxy, meaning that limited number of templates will force similarity for sequence modeled on the same templates.

The first problem is hard to addressed, but I would suggest discussing it. For example, the 3% public structures might become negligible if all possible antibodies from the repertoire are considered, and not only the ones for which modeling is possible.

As for the second problem, some template subsampling could give an idea of the effect of template paucity on the estimated diversity.

Minor

Introduction:

"V and J gene transcript origins and above a certain percentage (same length) Complementarity-Determining Re- gion H3 (CDRH3) sequence identity." -> the (same length) bit is unclear at this point, so remove it or expand it

Results

"41 Gidoni volunteers with sufficiently deep reads (see Methods)." -> with sufficient sequencing depth

Methods

- "Sequence clustering. The modellable chains were then se- quence clustered using CD-HIT (39) at a 90% sequence iden- tity threshold, to reduce the number of VH-VL pairing com- parisons to a computationally-tractable number."

I strongly advice against using CD-HIT for this kind of analyses. CD-HIT only assures that sequencing in the same cluster are >90% identical (even not so, since it doesnt properly take indels into consideration), but in large dataset, it will retain many cases where sequences that are >90% are in different clusters. This means that, the larger the initial dataset, the more likely 2 random representatives might actually be similar in sequence

-

**Have all data underlying the figures and results presented in the manuscript been provided?**

Reviewer #1: Yes

Reviewer #2: Yes

Reviewer #3: Yes

Reviewer #4: None

PLOS authors have the option to publish the peer review history of their article (what does this mean?). If published, this will include your full peer review and any attached files.

Reviewer #1: No

Reviewer #2: **Yes: **Jordan R Willis

Reviewer #3: No

Reviewer #4: No
---

## [Decision Letter · Decision Letter 1]

6 Jan 2021

Dear Charlotte,

I sincerely apologize for how long we have had this paper. We had strongly conflicting reviews and I was trying to get one more reviewer but that reviewer got COVID, so it has gotten complicated.

Two of the reviewers are in favor of publication while the other views the results as inconclusive, really because what you are trying to show is very hard to prove without more experimental data (e.g. paired VH/VL sequences, knowing which antibodies in different individuals bind a particular antigen). I tend to think the skepticism of this reviewer is warranted. But I think it is also fair to say that you have made a heroic effort to detect whatever signal can be detected and to respond to this reviewer's previous comments. I think at this point it makes more sense to follow through with publication and let readers decide how they view the data and its implications and significance. Certainly the quality of the analysis deserves publication in PLOS Computational Biology, even if different individuals would interpret it differently.

I have entered a decision of "Minor revisions" so that you can make any changes you want in response to the reviewers. [Form letter below]. But once you return the paper, we can proceed with acceptance.

Again, my apologies.

Best wishes,

Roland

Dear Prof Deane,

Thank you very much for submitting your manuscript "Public Baseline and Shared Response Structures Support the Theory of Antibody Repertoire Functional Commonality" for consideration at PLOS Computational Biology. As with all papers reviewed by the journal, your manuscript was reviewed by members of the editorial board and by several independent reviewers. The reviewers appreciated the attention to an important topic. Based on the reviews, we are likely to accept this manuscript for publication, providing that you modify the manuscript according to the review recommendations.

Sincerely,

Roland L. Dunbrack Jr., Ph.D.

Associate Editor

PLOS Computational Biology

Feilim Mac Gabhann

Editor-in-Chief

PLOS Computational Biology

[LINK]

Reviewer's Responses to Questions

**Comments to the Authors:**

Reviewer #1: The reworked version of the manuscript offers additional analyses, clarifies many points raised during the initial review and more precisely describes many of the claims offered by the available evidence. I would also like to acknowledge the immense amount of work that went into not only the original version of the manuscript but also in preparing these revisions. In my initial review of this work, I felt that the claims being made were too sweeping; the adjustments made by the authors have resulted in a much more nuanced and clear report. However, the addition of this nuance and clarity reveals that the central claim – that analyzing antibodies in terms of structure of the Fv instead of only considering shared genetic lineage among antibodies will yield a larger set of antibodies identified as similar to one another – is essentially tautologically true. Coupling that with the various sources of modeling errors and the inability to determine which of the structurally similar antibody models actually target the same epitopes leads me to feel that this work does not meet the high standards for publication in PLOS Computational Biology.

Major concerns:

1) One question that came to mind as I read this paper is the extent to which the computationally-expensive step of generating structural models provides a benefit over less expensive sequence-based approaches. If different-length CDR loops that could lead to similar paratopes (a task that is made extremely challenging by the inability to produce pairs of heavy and light chains that are known to form the donor’s functional antibodies) aren’t being considered, could an analysis following a sequence-based assignment to canonical CDR cluster types and the same 20-residue orientation prediction provide the same information as a complete structural model? The observation that longer CDR 3 loops lead to an overestimate of the degree of similarity between antibodies suggests considering only the length of the CDR loops can lead to template selection-based modeling errors that may be present elsewhere (and possibly inflating the number of public structures).

2) In the response to reviewers, the authors present a compelling case for ignoring the organism assignment from the PDB, but the case for combining template structures across organisms (with the possible exception of CDR H3) remains thin. While the antibody structures in the PDB are heavily engineered in such a way as to make the organism assignment uninstructive, the source of the frameworks and each CDR can be identified separately (e.g. with an HMM) and then grouped by source organism as opposed to PDB organism label. Does this procedure produce better results?

3) In the proximity to therapeutics section, the authors state “Of the 66 therapeutics with known structures that had at least one antibody in our ‘Public Baseline AML’ with 6 identical CDR lengths, all had a structural partner in the AML within a Cα Fv RMSD of 1.84Å…” Similar to the above concerns, how does this value change if one performs a sequence-based assignment to canonical clusters instead of only considering the length? While I would expect the number of antibodies under consideration to decrease, I would expect the maximal RMSD to decease as well (which would suggest that building structural models are not providing a benefit over a more robust sequence-based analysis).

4) In the same section, it is unclear how RSPs would aid in the design for screening libraries – I would recommend adding a description of the value they would add over in in addition to widely used antibody campaigns and how the major limitations of those techniques (including immunogenicity) would be effectively reduced. For example, when building a phage display library as suggested in the discussion, each mutation toward binding a particular target would essentially reset the question of immunogenicity and would require complete in vivo exploration for confirmation, thus neutralizing the advantage of developing the library from AMLs.

5) I appreciate the adjustment of the title of this paper. However, upon further consideration of the uncertainty in VH/VL pairing, the limitation of which sequences are considered modellable, the revealed biases associated with template availability and selection, and the uncertainty of the utility of the structural models over other sequence-based approaches (i.e. canonical cluster assignment and comparison), I feel it is too strong. While I agree with the overall thrust of this paper – that thinking in terms of structure of the Fv will almost certainly suggest a larger degree of functional commonality compared to only considering shared genetic lineage among antibodies – the previously listed limitations to generating models with atomic accuracy severely complicates the degree to which support for a particular theory can be extracted from them. I would recommend further adjusting the title to “Public Baseline and Shared Response Structures are Consistent with the Theory of Antibody Repertoire Functional Commonality”.

Reviewer #2: none

Reviewer #3: I am satisfied with the response to my queries, I think the authors have made a substantial effort in addressing all the criticisms.

**Have all data underlying the figures and results presented in the manuscript been provided?**

Reviewer #1: Yes

Reviewer #2: Yes

Reviewer #3: Yes

PLOS authors have the option to publish the peer review history of their article (what does this mean?). If published, this will include your full peer review and any attached files.

Reviewer #1: No

Reviewer #2: **Yes: **Jordan R Willis

Reviewer #3: **Yes: **Franca Fraternali
---

## [Editor Report · Decision Letter 2]

8 Feb 2021

Dear Prof Deane,

We are pleased to inform you that your manuscript 'Public Baseline and Shared Response Structures Support the Theory of Antibody Repertoire Functional Commonality' has been provisionally accepted for publication in PLOS Computational Biology.

Thank you for your considerable patience during this review. Your responses and additional work have made the paper stronger.

Best regards,

Roland L. Dunbrack Jr., Ph.D.

Associate Editor

PLOS Computational Biology

Feilim Mac Gabhann

Editor-in-Chief

PLOS Computational Biology

---

## [Editor Report · Acceptance letter]

24 Feb 2021

PCOMPBIOL-D-20-00654R2 

Public Baseline and Shared Response Structures Support the Theory of Antibody Repertoire Functional Commonality

Dear Dr Deane,

I am pleased to inform you that your manuscript has been formally accepted for publication in PLOS Computational Biology. Your manuscript is now with our production department and you will be notified of the publication date in due course.

With kind regards,

Alice Ellingham
